

# Simultaneous rapid detection of Hantaan virus and Seoul virus using RT-LAMP in rats

Xin Sui[1,2,*], Xu Zhang[1,3,*], Dongliang Fei[1], Zhen Zhang[3] and Mingxiao Ma[1]

[1] Institute of Biological Sciences, Jinzhou Medical University, Jinzhou, Liaoning, China
[2] The First Affiliated Hospital, Jinzhou Medical University, Jinzhou, Liaoning, China
[3] Microbiological Laboratory, Center for Disease Control and Prevention of Jinzhou, Jinzhou, Liaoning, China
* These authors contributed equally to this work.

Corresponding author
Mingxiao Ma,
mamingxiao@jzmu.edu.cn

## ABSTRACT

**Background:** Hemorrhagic fever with renal syndrome is in most cases caused by the Hantaan virus (HTNV) and Seoul virus (SEOV). To develop and apply reverse transcription loop-mediated isothermal amplification (RT-LAMP) to detect HTNV and SEOV simultaneously, which was faster, more cost effective, and easier to perform as the target gene amplified rapidly. In this article an assay based on LAMP is demonstrated, which only employs such apparatus as a water bath or a heat block.
**Methods:** A chromogenic method using the calcein/$Mn^{2+}$ complex and real-time turbidity monitoring method were used to assess reaction progress of the reaction, and the specificity of the RT-LAMP-based assay was assessed by detecting cDNAs/cRNAs generated from Coxsackievirus A16, Influenza virus, lymphocytic choriomeningitis virus, mouse poxvirus, rotavirus, mouse hepatitis virus. In addition, 23 clinical specimens were used to determine the agreement between the RT-LAMP assay with reverse transcriptase polymerase chain reaction (RT-PCR) and immunofluorescence (IFT) method.
**Results:** The detection limit of RT-LAMP to HNTV and SEOV was as low as 10 copies/µL with optimized reaction conditions, which was much more sensitive than the RT-PCR method (100–1,000 copies/µL). At the same time, the detection results of 23 clinical specimens have also illustrated the agreement between this the RT-LAMP assay with RT-PCR and IFT.
**Discussion:** This RT-LAMP assay could be used to perform simultaneous and rapid detection of HTNV and SEOV to the clinical specimens.

## INTRODUCTION

Hemorrhagic fever with renal syndrome (HFRS) is characterized by fever, acute kidney damage, and hemorrhage in the clinical symptoms (*Schmaljohn & Dalrymple, 1983*; *Zou, Chen & Sun, 2016*; *Liu et al., 2007*). Most cases have been reported from mountainous areas, with about 200,000 deaths worldwide annually (*Jonsson, Figueiredo & Vapalahti, 2010*;
*Kim et al., 2016b*). China is the most important endemic regions of HFRS, which reach to 90% of total cases worldwide (*Zou, Chen & Sun, 2016*; *Li et al., 2016*). At present, HFRS is regarded as one of top nine communicable diseases in mainland China (*Zou, Chen & Sun, 2016*). At the same time, HFRS has been detected in all 31 provinces of China (*Zou, Chen & Sun, 2016*). Research have shown that HFRS is caused by hantaviruses (HVs) within the Bunyaviridae family, belonging to negative sense RNA viruses (*Kim et al., 2016b*). In Asia and Europe, four main sero/geno types of HFRS-associated HVs have been identified, including Seoul virus (SEOV), Hantaan virus (HTNV), Dobrava-Belgrade virus, and Puumala virus, and among these, HTNV and SEOV are the major pathogen of HFRS in China (*Kim et al., 2016a*; *Zhang et al., 2014*; *Zhang et al., 2009*). HTNV RNA genome contains three gene segments: L (large), M (medium), and S (small), which encode different proteins, including RNA-dependent RNA polymerase (L), glycoproteins (Gn and Gc), and nucleocapsid protein (N) (*Kim et al., 2016b*). Due to serious threat to public health, it is exceedingly useful that an accurate and efficient detection method of HTNV was establishment, which would make more opportunity for the patients to receive treatment in the early stages of HFRS disease.

Currently, the traditional diagnosis methods to HFRS are mainly depended on exposure history, typical clinical symptoms, immunofluorescence (IFT) and serological experiment, such as IgM or IgG antibody level against HTNV by enzyme-linked immunosorbent assay. However, the traditional serological methods can't assess the replication of the virus in patient's blood, and there is some cross-reactivity in different HVs present (*Lundkvist et al., 1997*; *Vaheri, Vapalahti & Plyusnin, 2008*). Molecular diagnostic methods, such as the reverse transcriptase polymerase chain reaction (RT-PCR) and quantitative RT-PCR (qRT-PCR) assays, have been established in HV detection (*Jiang et al., 2014*). However, these PCR-based assays require expensive instruments, specialized technicians, and complicated procedures, so it is unsuitable for rapid diagnostics in field situations.

The loop-mediated isothermal amplification (LAMP) assay is a novel method by the rapid, high sensitivity, and isothermal nucleic acid amplification, which relies on Bst DNA polymerase in conjunction with two inner primers and two outer primers (which recognize six specific areas of the target gene) to amplify target gene at 60–65 °C within 60 min (*Notomi et al., 2000*). Since the establishment of LAMP assay in 2000, this method has been widely applied to the detection of various pathogens (*Mori & Notomi, 2009*; *Seki et al., 2018*; *Guo et al., 2018*; *Zheng et al., 2018*), and the LAMP assay have also been used as the rapid diagnosis method of HFRS based on blood specimens of the patients (*Hu et al., 2015*), but blood collection isn't convenient to the rodents in the wild, and it is necessary to develop the new rapid diagnosis method of HFRS that fit for detecting the rodents in the field.

This paper reports on the establishment of a rapid detection method for HTNV and SEOV by reverse transcription loop-mediated isothermal amplification (RT-LAMP), and the S segment sequences of HTNV and SEOV were used as a target gene to be amplified in vitro transcription. This RT-LAMP assay of HTNV and SEOV has been

shown to be more rapid, efficient and timesaving with compared to the traditional serological methods. Moreover, the RT-LAMP assay has similar specificity and sensitivity with RT-PCR and qRT-PCR assays, and this technique provides a novel and effective diagnostic method for HTNV and SEOV.

# MATERIALS AND METHODS

## Ethics statement

This research was approved by the Experimental Animal Ethics Committee of Jinzhou Medical University (Approval: 20150918).

## Samples

The purified HTNV strain RH153 and SEOV strain RH164 were gifts from Professor Liu Xuesheng of Liaoning Province center for disease control and prevention (CDC), and stored at −80 °C; lymphocytic choriomeningitis virus (LCMV), mouse poxvirus (MPV), rotavirus (RV), and mouse hepatitis virus (MHV)-infected rat lungs were donated by Professor Zhengming He of the National Institutes for Food and Drug Control (China), and influenza virus (IV), Coxsackievirus A16 (CA16), and lungs of healthy rat were preserved in Jinzhou (CDC), and stored in a liquid nitrogen; The clinical lung sample were collected from the captured rat in the Heishan, Guta, Beizhen, and Taihe of Jinzhou during 2008–2013, which were trapped with rat traps or baited cages in the tussocks, canals and fields, then stored in a liquid nitrogen until further processing.

## Primers design

By comparison with HTNV and SEOV stains of China, S gene of SEOV strain Rn-HD164 (GenBank: GQ279392.1) was used as the reference sequence to design the primers of RT-LAMP (Figs. S1, S2 and Word S1). In order to ensure the specificity of the assay, six primers were designed to identify six regions of the S target gene, and five set RT-LAMP primers for HTNV and SEOV were designed based on the S gene sequence of Rn-HD164 strain using the website (http://primerexplorer.jp/e/).

Each set of primers contained two inner primers (FIP and BIP), two outer primers (F3 and B3), and two loop primers (LB and LF). The primer sequences used in the RT-PCR assays have been referred to National Health and Family Planning Commission of the People's Republic of China. Primers were synthesized by Shanghai Sangon Biotech (Shanghai, China), and the sequences and positions of the RT-LAMP and RT-PCR primers are shown in Table 1.

## RNA extraction

Lung tissue (about four mm$^3$) from sample rats were obtained antiseptically in the Class II biological safety cabinet, then total RNA was extracted using the Qiagen RNAmini Kit (Qiagen, Venlo, Netherlands) according to the manufacturer's directions. cDNA was synthesized with Oligo dT-Adaptor Primer and first-strand cDNA synthesis kit (Invitrogen, Carlsbad, CA, USA). The contents and volume of reaction mixture were as follows: MgCl$_2$ two µL, 10×RT Buffer one µL, RNase Free dH$_2$O 3.75 µL, dNTP Mixture (10 mM) one µL, RNase Inhibitor 0.25 µL, AMV Reverse Transcriptase 0.5 µL,

**Table 1 The primers used for RT-LAMP and RT-PCR.**

| Set | Reaction type | Primer number | Position | Sequence(5′–3′) |
|---|---|---|---|---|
| I | RT-LAMP | HS-72F3 | 388–405 | ACAGCTGATTGGTTGACT |
| | | HS-72B3 | 584–601 | TTGATTGGGCATTTGGCA |
| | | HS-72FIP | 406–430, 453–477 | ACCTCTTGTTGTTAACATGTACAGTTTATAATTGTCTATCTGACATCATTCG |
| | | HS-72BIP | 509–531, 563–583 | GGATCAGATTCAAGGATGACAGCTTTTGACACATACAGATGTTTGG |
| | | HS-72LF | | GCCTTCAAGATGATTGGGACCA |
| | | HS-72LB | | TGAGGATGTCAATGGAATCAGAAAG |
| II | RT-LAMP | HS-154F3 | 647–666 | TATGTGGGTTATATCCTGCA |
| | | HS-154B3 | 826–847 | TCTGTCTGATATAGTCACGATT |
| | | HS-154FIP | 709–728, 669–689 | GCCAGTGCCAAAAACCCAACTTGATAAAGGCAAGGAACATGGT |
| | | HS-154BIP | 743–766, 803–822 | CTAGAATTGAAGAATGGCTTGGCGTTAGGATTCCCAGATAAACTCC |
| | | HS-154LB | | CACCCTGCAAGTTCATGGCAGAG |
| III | RT-LAMP | HS-203F3 | 803–822 | GGAGTTTATCTGGGAATCCT |
| | | HS-203B3 | 976–993 | TGGTGGACACCTATCAGG |
| | | HS-203FIP | 824–848, 864–886 | GAAATTCCTTTGGCTCCATTCCTTTTAAATCGTGACTATATCAGACAGAG |
| | | HS-203BIP | 945–963, 892–911 | CTCAGGCAACATGCAAAGGATTCACCCATATTGACGATGGT |
| | | HS-203LB | | TGCTGGATGTACACTGGTTGA |
| IV | RT-LAMP | HS-1F3 | 15–39 | CTAAAGAGCTATTACACTAACAAGA |
| | | HS-1B3 | 215–232 | GGCGCTTCAATTCATCAA |
| | | HS-1FIP | 85–104, 44–64 | GCTATCACAAGCTGCCCCTCTTTGGCAACTATGGAAGAAATCC |
| | | HS-1BIP | 121–145, 183–200 | GATGCAGAAAGCAGTATGAGAAGGTTGAAGCTGCAACACTCTCC |
| | | HS-1LB | | ATCCTGATGACTTAAACAAGAGGG |
| V | RT-LAMP | HS-124F3 | 539–559 | AGGATGTCAATGGAATCAGAA |
| | | HS-124B3 | 702–721 | CCAAAAACCCAACTACACTC |
| | | HS-124FIP | 601–625, 561–580 | GTGTTATCTCTTCAGCTTTCATGCTTTGCCCAAACATCTGTATGTGT |
| | | HS-124BIP | 629–648, 684–701 | GAAGATTCCGCACGGCAGTATTATGACAGGGCTTACCATG |
| | | HS-124LB | | TGTGGGTTATATCCTGCACAGA |
| VI | RT-PCR | HHSIF | 688–709 | GTAAGCCCTGTCATGAGTGTAG |
| | | HHSIQ | 885–907 | TTGCATGTTGCCTGAGGGCTTGA |

**Note:**
The gene region used for designing primers from the S gene (GenBank: GQ279392).

Oligo dT-Adaptor Primer 0.5 μL which made up to 10 μL. The reaction conditions were as follows: 30 °C for 10 min, 42 °C for 30 min, 50 °C for 30 min, 95 °C for 15 min, and 5 °C for 5 min. The cDNA was stored at −20 °C. Viral RNA was extracted using the TIANamp Virus DNA/RNA Kit (Tiangen, Beijing, China) according to the manufacturer's instructions.

## RT-LAMP assays

The LAMP reaction was performed in 25-μL volume which contained 20 mM Tris-HCl (pH 8.8) (Shanghai Bioscience Biological Technology Company, Shanghai, China), 10 mM KCl and 10 mM $(NH_4)_2SO_4$ (Sinopharm Chemical Reagent Co., Shanghai, China), 0.1% Triton X-100 (Solebo Biotechnology Corporation, Beijing, China), 0.8M lysine (Sigma-Aldrich, St. Louis, MO, USA), eight mM $MgSO_4$ (Sinopharm Chemical Reagent Co.,

Shanghai, China), 1.4 mM dNTP, eight U of Bst DNA polymerase and reverse transcriptase, 40 $p$mol of each primer FIP and BIP, five $p$mol of each primer F3 and B3, and 20 $p$mol of each primer LB and LF. The reaction was performed at 65 °C for 60 min, and double-distilled water was used as a negative control.

Loop-mediated isothermal amplification products were analyzed by two independent methods, either sample turbidity or fluorescence. The LA-320 turbidimeter (Eiken Chemical Co., Ltd, Tokyo, Japan) was used to monitor the turbidity of RT-LAMP products in real time (measured at 6-s intervals on 650 nm wave length), and the samples were considered positive if the value of sample turbidity exceeded 0.1. The turbidity changes arose from the presence of LAMP reaction by-product a white precipitate ($Mg_2P_2O_7$). The second method used visual inspection to assess color changes in the presence of the fluorescent metal ion indicator calcein/$Mn^{2+}$ complex, and the samples that turned from orange to green on reaction completion, while no color change samples, were considered negative.

## Optimization of primers and temperature for RT-LAMP assay

To determine the optimal primer for RT-LAMP assay for HTNV and SEOV detection, five primer sets were detected in the same reaction condition using real-time turbidimeter LA-320 and their turbidity curves were draw at 650 nm according to the amplified results. Then the optimal primers were used to initiate RT-LAMP reaction at different temperatures ranging from 59 to 66 °C with a gradient of 1 °C.

## RT-PCR assays

Reverse transcriptase polymerase chain reaction was conducted in accordance with the manufacturer's protocol. Briefly, five μL of RNA sample was added to a 20 μL reaction mixture containing 100 ng Oligo dT (15mer), 1X AMV/Tfl buffer, one mmol/L $MgSO_4$, 10 mmol/L DTT, 200 μmol/L of each dNTP, 300 nM each specific primer, 2.5 units each of AMV Reverse Transcriptase and Tfl DNA Polymerase and 20 U RNasin. RT-PCR assays was followed uninterrupted thermal cycling programs consisting of 45 min at 42 °C, 4 min at 94 °C, 35 cycles of 30 s at 94 °C, 30 s at 55 °C and 30 s at 72 °C and a final elongation step of 10 min at 72 °C. RT-PCR products were electrophoresed in 1% agarose gel and visualized using UV light after ethidium bromide staining.

## Sensitivity comparison of RT-LAMP and RT-PCR

For comparison of sensitivity between RT-LAMP assay and RT-PCR, the positive controls (cRNA templates of the purified HTNV strain RH153 and SEOV strain RH164) were detected on the same samples. After the cRNA template concentration of the purified HTNV and SEOV was measured using the NanoDrop One microvolume UV–Vis spectrophotometer (ThermoFisher, Waltham, MA, USA), the viral copy numbers were calculated from the full length genome using the formula (copies/μL = ng/μL × Avogadro constant/1 bases of average molecular mass × genome length), then seven assay standards were prepared by 10-fold serial dilutions containing $10^5$–$10^{-1}$ copies/μL, respectively. The cRNA of the healthy rat lung tissue was used as the negative control. The colorimetric method (the addition of calcein) and the turbidity method (determination of turbidity)
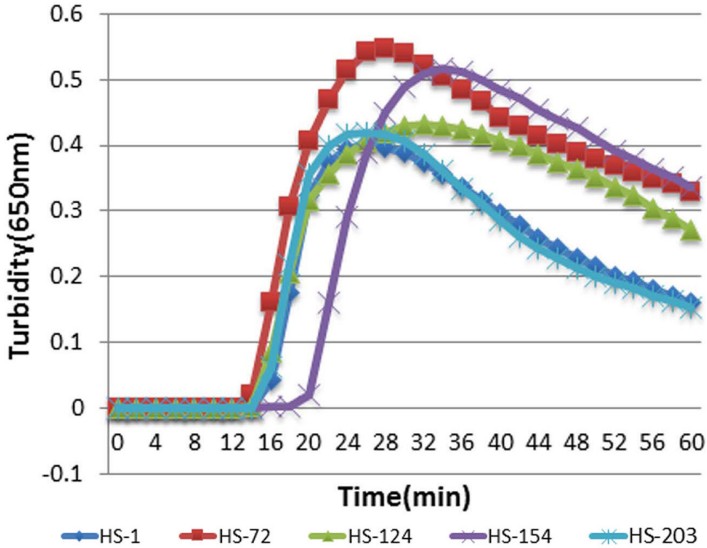

**Figure 1** **The results of RT-LAMP with different sets of primers.** Each set of primers was tested with the same concentration of templates. Turbidity indicated the level of amplification. The reaction mixture with HS-72 achieved a turbidity of approximate 0.06 in 15 min, while others were below 0.017 at this point.                                                              

were both used to monitor the reaction of LAMP. At the same time, the RT-PCR products were electrophoresed in 1% agarose gel and visualized using UV light.

## Specificity test of RT-LAMP
The specificity of RT-LAMP was assessed by testing cDNAs generated from CA16, IV, LCMV, MPV, RV, MHV, lungs of healthy rats and HTNV, SEOV positive samples. Distilled water was used as negative control.

## Examination of clinical samples
A total of 23 rat lung samples were randomly selected from those collected between 2008 and 2013. All samples were collected according to the national guideline for prevention and treatment of epidemic hemorrhagic fever which states that night trapping method was used to seize rats randomly in the countryside and residential area (Zhang et al., 2004). Cordotomy was used to kill all the rats. Each of them was tested with RT-PCR, IFT, and RT-LAMP.

## RESULTS
### Optimization of primers
To determine the optimal primers for HTNV and SEOV, five primers were detected in the same reaction condition using real-time turbidimeter L320C and their turbidity curves were screened at 650 nm according to the amplified results. As illustrated in Fig. 1, HS-72 was selected as optimal primers (Table 1).

### Determination of optimal reaction temperature
The optimal primers HS-72 were used to initiate RT-LAMP reaction at different temperatures ranging from 59 to 66 °C with a gradient of 1 °C. As shown in Fig. 4,

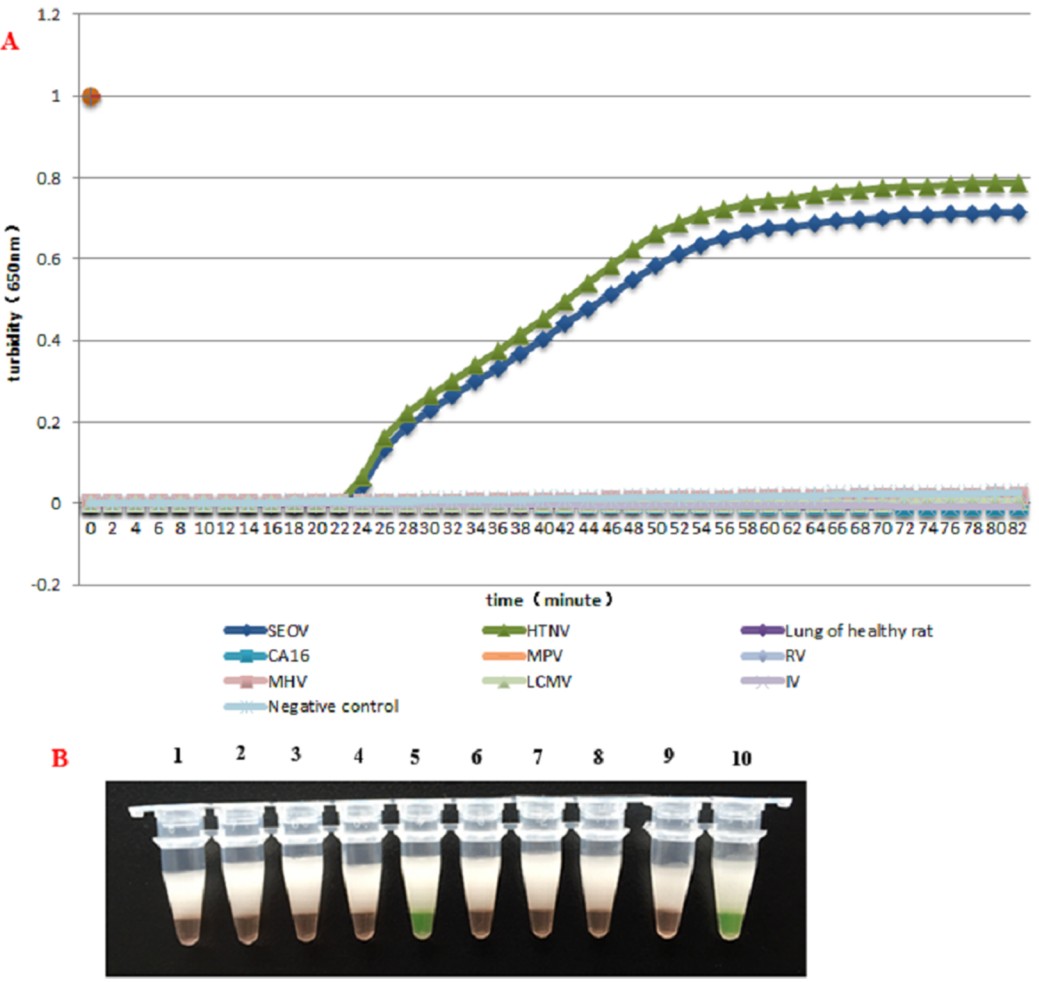

**Figure 2** **The specificity assay of RT-LAMP.** (A) Results were analyzed using an LA-320C turbidimeter: Amplification was carried out with the same reaction mixture and the same concentration of different templates, and only the reaction mixture with HTNV and SEOV samples showed an amplification curve, and no amplification was found with other samples. (B) Results were analyzed through color changes: Tube (1), Lung of healthy rat; (2), Negative control; (3), CA16; (4), LCMV; (5), HTNV; (6), IV; (7), MPV; (8), MHV; (9), RV; (10), SEOV.

RT-LAMP was first initiated at 64, 65, and 66 °C. However, the higher temperature could have an effect on the activity of Bst DNA polymerase and the reaction time was longer with temperature increasing, so 65 °C was selected as the optimal temperature.

## Specificity test

HS-72 was used with all the provided samples in subsequent reactions. HS-72 displayed high specificity and could be used for simultaneous detection of HTNV and SEOV. Only HTNV and SEOV were detected with HS-72 and none of the samples with other viruses were diagnosed as positive, suggesting a good specificity (Figs. 2A and 2B).

## Sensitivity test and comparison with PCR

For templates at concentrations from $1 \times 10^5$ to $1 \times 10^{-1}$ copies/L, RT-LAMP assays were carried out by the LA-320C turbidimeter. As illustrated in Figs. 3A–3D, both the turbidity

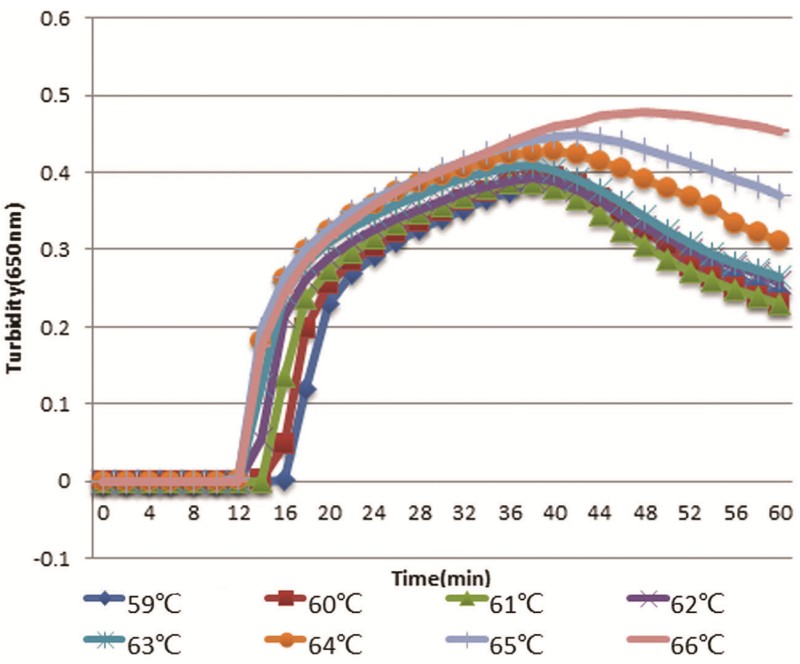

**Figure 3 Optimal temperature assay of RT-LAMP.** Turbidity indicated the level of amplification. The reaction at 65 °C showed a turbidity of 0.044 at as early as 12 min, while the turbidity of other reactions was below 0.001 at this time.

and color detection limit of the RT-LAMP assay was up 10 copies/μL for HTNV and SEOV. Comparison to the RT-LAMP assay, the RT-PCR limits of detection for HTNV and SEOV were $10^2$ and $10^3$ copies/L, respectively (Figs. 3E and 3F).

## Test results of clinical samples

The 23 clinical samples were detected to determine the feasibility of the RT-LAMP assay, and result have shown in Figs. 5A and 5B. At the same time, all the samples were subjected to conventional RT-PCR and IFT. In the result, the RT-LAMP method showed that five samples were determined as the positive of SEOV and HTNV, and four positive specimen in RT-PCR and only three in the IFT assay, which was superior to RT-PCR and IFT methods (Table 2). From shown in Table 2, all three methods yielded positive results for the JZHS-5, LZLH-4, JZGT-1 samples, and JZGT-2 sample wasn't detected using RT-PCR, but the detection of RT-LAMP were positive.

## DISCUSSION

Since its development, LAMP has been widely applied in diagnosis of clinical illness, quantitative and qualitative detection of epidemic bacteria and viruses, and gender determination of animal embryos (*Mori & Notomi, 2009*; *Seki et al., 2018*; *Guo et al., 2018*; *Zheng et al., 2018*; *Hirayama et al., 2006*). HVs are among the most important zoonotic pathogens of humans and the subject of heightened global attention (*Guo et al., 2013*), and previous studies show that rodents have long been recognized as the principal reservoirs of HVs (*Ermonval, Baychelier & Tordo, 2016*), so it is important to develop

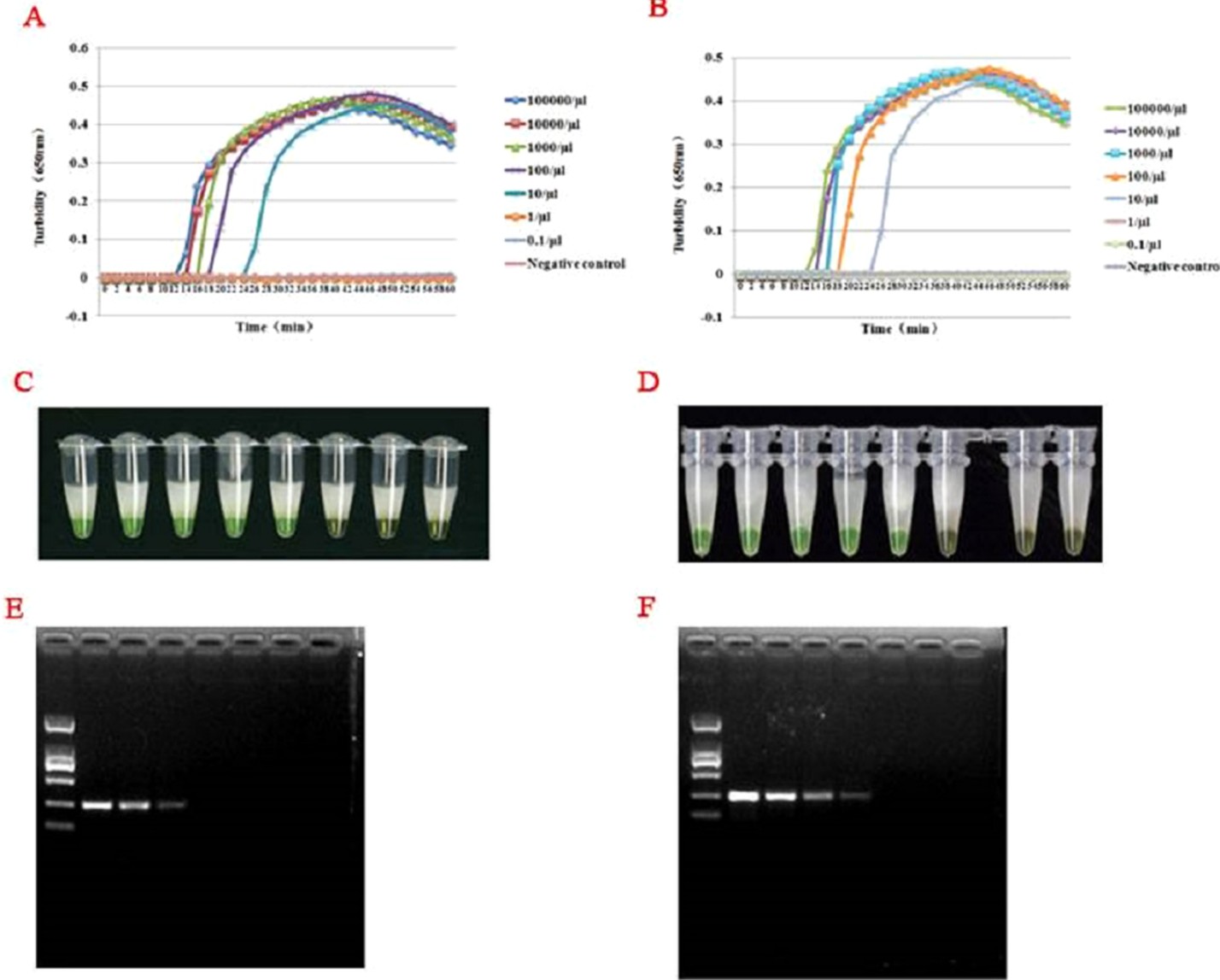

**Figure 4 Comparison of hantavirus detection sensitivity by RT-LAMP and RT-PCR assays.** Results were analyzed using an LA-320C turbidimeter (A and B): both HTNV and SEOV were successfully detected with a detection range of 10–100,000 copies/µL, but the samples with a concentration of one copy/µL and 0.1 copy/µL wasn't detected, and the negative control showed no amplification either. Color formation indicated the level of amplification (C and D). From right to left: $10^5$ copies/µL, $10^4$ copies/µL, $10^3$ copies/µL, $10^2$ copies/µL, $10^1$ copies/µL, $10^0$ copies/µL, $10^{-1}$ copies/µL, and negative control. A significant difference in reaction color was observed in presence of amplification (the left five green tubes) and in absence of amplification (the three yellow ones on the right). Results of the sensitivity assay using RT-PCR (E and F): From right to left: $10^5$ copies/µL, $10^4$ copies/µL, $10^3$ copies/µL, $10^2$ copies/µL, $10^1$ copies/µL, one copies/µL, $10^{-1}$ copies/µL, negative control. DNA marker: D2000. HTNV and SEOV could be detected at as low as 100–1,000 copies/µL. 

effectively measures to investigate HFRS infection in rodents in the wild. Although LAMP have been used as the rapid diagnosis method of HFRS based on blood specimens of the patients (*Hu et al., 2015*), but the sensitivity for HTNV was 10-fold lower than that for SEOV in this assay which probably led to undetection for HTNV, and the blood collection isn't convenient to the rodents in the wild, so the method was unsuitable to rapid
**A**

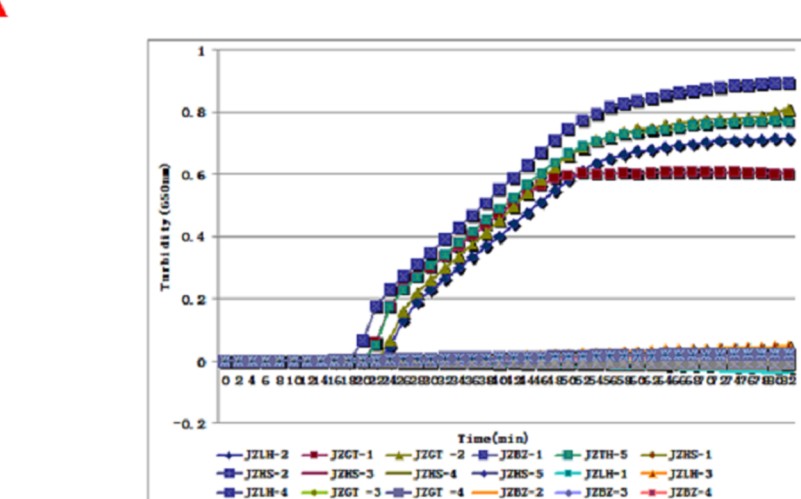

**B**

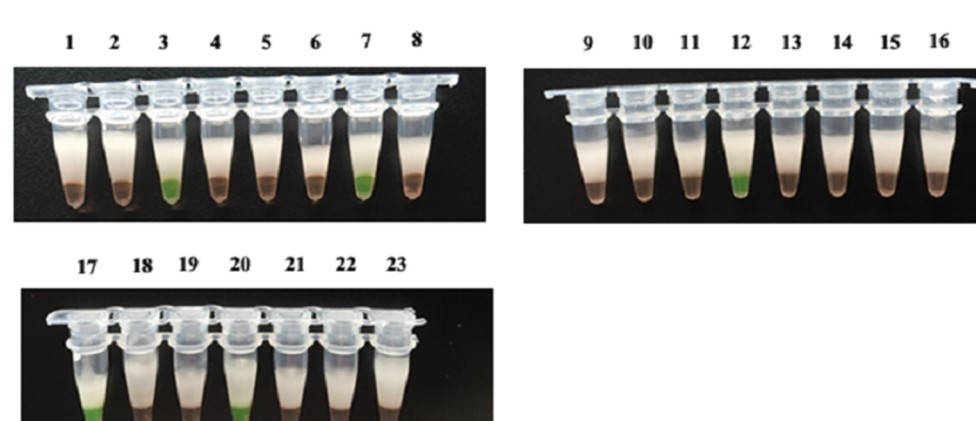

**Figure 5 Detection of clinical specimens by RT-LAMP assay.** (A) Results were analyzed using the LA-320C turbidimeter: JZGT-2, JZTH-5, JZGT-1, JZLH-4, and JZHS-5 were determined to be positive in the 23 clinical specimens. (B) Results were analyzed through color changes: (1–8): JZHS-1, JZHS-2, JZGT-2, JZLH-1, JZLH-2, JZLH-3, JZTH-5, JZHS-3; (2–16): JZHS-4, JZGT-3, JZGT-4, JZGT-1, JZBZ-1, JZBZ-2, JZBZ-3, JZBZ-4; (17–23): JZLH-4, JZBZ-5, JZTH-1, JZHS-5, JZTH-2, JZTH-3, JZTH-4.

epidemiological investigation of HVs (*Ermonval, Baychelier & Tordo, 2016*). In this research, specific primers for HTNV and SEOV were designed to carry out RT-LAMP using lung tissue specimens, and the new method is more suitable to apply in the wild. At the same time, the new RT-LAMP assay not only showed high sensitivity and rapidity, but also could be finished in less than 1 h with only a heated water bath required, so this method appears to be more suitable in the endemic areas with poorly equipped laboratories. Currently, the detection cost to each sample of RT-LAMP assay is about three times as compared to RT-PCR method, but the cost will be remarkably reduced as the extensive application. Furthermore, most researchers recommend electrophoresis of amplification products or addition of SYBR Green I fluorochrome to the products as a

**Table 2 Summary of test results of 23 clinical specimens.**

| Samples | Area of collecting sample | RT-PCR | RT-LAMP | IFT |
|---|---|---|---|---|
| JZHS-1 | Heishan | − | − | − |
| JZHS-2 | Heishan | − | − | − |
| JZHS-3 | Heishan | − | − | − |
| JZHS-4 | Heishan | − | − | − |
| JZHS-5 | Heishan | + | + | + |
| JZLH-1 | Linghai | − | − | − |
| JZLH-2 | Linghai | − | − | − |
| JZLH-3 | Linghai | − | − | − |
| JZLH-4 | Linghai | + | + | + |
| JZGT-1 | Guta | + | + | + |
| JZGT-2 | Guta | − | − | + |
| JZGT-3 | Guta | − | − | − |
| JZGT-4 | Guta | − | − | − |
| JZBZ-1 | Beizhen | − | − | − |
| JZBZ-2 | Beizhen | − | − | − |
| JZBZ-3 | Beizhen | − | − | − |
| JZBZ-4 | Beizhen | − | − | − |
| JZBZ-5 | Beizhen | − | − | − |
| JZTH-1 | Taihe | − | − | − |
| JZTH-2 | Taihe | − | − | − |
| JZTH-3 | Taihe | − | − | − |
| JZTH-4 | Taihe | − | − | − |
| JZTH-5 | Taihe | + | − | + |

monitoring method. However, both these methods expose the products to the atmosphere, thus increasing the risk of false positive results. Additionally, traditional electrophoresis methods fail to perform real-time monitoring. Our study employs real-time nephelometry and spectrophotometry without opening the reaction tube. Therefore, contamination is avoided, thereby preventing false positive outcomes.

Reverse transcription loop-mediated isothermal amplification showed better potential than PCR in many aspects. Significantly, RT-LAMP featured isothermal amplification, which is independent of the thermal cycler system, thus promoting application of LAMP. It is known that addition of lysine facilitates isothermal amplification in presence of Bst DNA polymerase by achieving dynamic equilibrium between DNA denaturation and renaturation. Further, *Notomi (2007)* found that the existence of impurities and huge amounts of exogenous DNA hardly affected the RT-LAMP reaction. It has been shown that six copies of HBV DNA could be detected by the LAMP reaction without being interfered with by 100 ng of human DNA. In a study by *Kaneko et al. (2007)*, extraction of DNA from the sample was required due to the insusceptibility of LAMP to other components in the samples. However, PCR can be disturbed by the presence of exogenous DNA and inhibitors in the samples. Hence, LAMP is more suitable for clinical samples

than PCR. In addition, the specificity of the LAMP reaction was much higher on account of its specific recognition of six distinct regions by four primer pairs (*Notomi et al., 2000*). In contrast, only two distinct regions are recognized in PCR. Therefore, the false positive rate of LAMP was largely lowered. In our study, a combination of optimal primer pair HS-72 was selected. The optimal concentration of each set of primers was as follows: 40 *p*mol FIP and BIP, five *p*mol F3 and B3, 20 *p*mol LB and LF. The optimal reaction temperature was 65 °C. The optimal reaction time was 60 min.

In this study, RT-LAMP was used for the rapid and simultaneous detection of HTNV and SEOV. The specificity test demonstrated that the target genes of HTNV and SEOV could be specifically amplified. The sensitivity detection demonstrated a detection limit of 10 copies/µL of HTNV and SEOV, and this assay is superior to traditional RT-PCR and IFT methods. In summary, RT-LAMP is suited for sensitive and rapid detection of HTNV and SEOV. Because of its simplicity and efficiency, this method is likely to be used to detect HTNV and SEOV in samples collected in the field. This will enable researchers to conduct epidemiological investigations that could provide the basis for prevention of HTNV and SEOV.

## CONCLUSIONS

The method for the detection of both HTNV and SEOV has been developed in this study, which is of great significance for the diagnosis of HFRS. With better sensitivity and comparable specificity to current diagnostic tests, and better availability in laboratories of various standards, this method has the potential to be utilized on a large scale.

### Funding
This work was supported by the Award for "Liaoning Distinguished Professor Award". The funders had no role in study design, data collection and analysis, decision to publish, or preparation of the manuscript.

### Grant Disclosure
The following grant information was disclosed by the authors:
Liaoning Distinguished Professor Award.

### Competing Interests
The authors declare that they have no competing interests.

### Author Contributions
- Xin Sui performed the experiments, prepared figures and/or tables, approved the final draft.
- Xu Zhang performed the experiments, analyzed the data, prepared figures and/or tables, authored or reviewed drafts of the paper, approved the final draft.
- Dongliang Fei analyzed the data, prepared figures and/or tables, approved the final draft.
- Zhen Zhang analyzed the data, prepared figures and/or tables, approved the final draft.

- Mingxiao Ma conceived and designed the experiments, analyzed the data, contributed reagents/materials/analysis tools, authored or reviewed drafts of the paper, approved the final draft.

## Animal Ethics

The following information was supplied relating to ethical approvals (i.e., approving body and any reference numbers):

The animal samples used in this research were collected according to the National Scheme for the prevention and Treatment of epidemic hemorrhagic fever. The trappers were permitted to trap the rats according to the National Scheme for Supervision of HFRS which was a multi-level scheme from country to city. This research was approved by the Experimental Animal Ethics Committee of Jinzhou Medical University.

## Data Availability

The raw data is available as Supplemental Files.

## Supplemental Information

Supplemental information for this article can be found online at http://dx.doi.org/10.7717/peerj.6068#supplemental-information.

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
