# Peer review of "Simultaneous rapid detection of Hantaan virus and Seoul virus using RT-LAMP in rats"

_PeerJ, doi:10.7717/peerj.6068_

## Round 0.1 · original submission · Major Revisions

I fully agree with both reviews that the manuscript requires major revisions. Particularly, the authors should cite previous studies and explain better the novelty of their work.

·

Basic reporting

The study proposed by Zhang et al entitled: Simultaneous rapid detection of Hantaan virus and Seoul virus using RT-LAMP in rats, is in line with the state-of-art in the literature since is focused on the development of an isothermal amplification system for the diagnosis. However the novelty of the assay is limited since other report with the same aim has been previously published (Hu et al.2015. J Virol Methods. 2015 Sep 1;221:68-73. doi: 10.1016/j.jviromet.2015.04.017.), In this regards I have found some critical point that I think the authors must address before to accept the manuscript.

Experimental design

Following the direction of the OIE manual and WHO manual for the validation of diagnostic assays based on amplification of nucleic acid the analytical sensitivity of the assay to be developed needs to be assessed in the same clinical samples where assay is expected to detect the agent. For this reason, I strongly recommend to the author that they must show the sensitivity of the new assay in clinical samples and not in distilled water, since this approach is not correct.

Validity of the findings

Some critical point are missing such as:
1-Analytical sensitivity determined on clinical material where the assay is addressed (not in distilled water)
2-Analytical specificity showing the assay can amplify hantavirus regardless the gentic diversity (representative member from the main four clades)
3-A analytical comparison between the new assay and the previously reported by Hu et al.2015. J Virol Methods. 2015 Sep 1;221:68-73. doi: 10.1016/j.jviromet.2015.04.017.

Additional comments

The study proposed by Zhang et al entitled: Simultaneous rapid detection of Hantaan virus and Seoul virus using RT-LAMP in rats, is in line with the state-of-art in the literature since is focused on the development of an isothermal amplification system for the diagnosis. However the novelty of the assay is limited since other report with the same aim has been previously published, In this regards I have found some critical point that I think the authors must address before to accept the manuscript.
Major comments:
1-Another RT-LAMP for simultaneous detection of Hanta virus and seoul virus has been published in 2015, an experimental comparison between both assays showing advantages and disadvantages of the new proposed assay is needed. This comparison will allow to the reader to understand why a new assay based on the same principle that the previously reported is needed in the scientific literature.
2-Te authors designed the primers based on GQ279392.1 GenBank strains however considering the genetic diversity of Hantavirus members four distinctive clades are described. Based on a vast number of reports about RT-LAMP assays, is well know that the hybridization of B3 and BIP primers are critical for the isothermal amplification. Therefore in order to guarantee that the assay really amplify these other clades of hantavirus the author must follow two directions: i) show at least in supplementary information with an alignment of sequences including sequences from the diverse clades of Hantavirus members others than the used for the primers design, ii)use in the analytical specificity of the assays representative strains from these clades, to guarantee that not only in silico but in vitro the assay can specifically amplify diverse genetic background into the Hantavirus.
3-Following the direction of the OIE manual and WHO manual for the validation of diagnostic assays based on amplification of nucleic acid the analytical sensitivity of the assay to be developed needs to be assessed in the same clinical samples where assay is expected to detect the agent. For this reason, I strongly recommend to the author that they must show the sensitivity of the new assay in clinical samples and not in distilled water, since this approach is not correct.
Considering the elements provided above I am willing to review a new version of the manuscript with the needed amends, however in the current form I can not accept the manuscript for its publication.

Reviewer 2 ·

Basic reporting

The paper is written in generally correct language, but somewhat confusing, with figures being called out of numerical order in the text, and some important issues are touched all too briefly, and should deserve more attention for the sake of clarity.

Literature review has important gaps. It is important that authors consider and discuss results previously published by others, based on studies that were conducted in the same country. For example, a very careful study was published by Hu D et al. in 2015, on the "Development of reverse transcription loop-mediated isothermal amplification assays to detect Hantaan virus and Seoul virus" (Journal of Virological Methods 221: 68–73, 2015). The lack of mention to that study is suggestive of carelessness in the literature review. The sensitivity of that assay was 10 copies (similar to PCR) for HTNV, and 100 copies for SEOV (ten-fold less than qPCR). These comparisons deserve mention.

Legends for figures are poor and some are impossible to understand entirely for lack of explanation.

Experimental design

Methods are insufficiently described.

Validity of the findings

The information is important, but it is still unclear how it relates to previous very similar studies.

Additional comments

This paper addresses the very important issue of developing sensitive and specific methods for the direct detection of the Old World hantaviruses Hantaan and Seoul in rats. However, the authors failed to make clear the reasons why their test is better than others, and a more accurate focus on that issue is needed.
Certain other issues deserve clarification.
Major concerns:
1-What hand-foot-and-mouth causing virus was used? Coxsackie? What type? (line 99)
2-How were the lungs of infected rats obtained? It is not enough to mention that they came to the authors from the Center for Disease Control and Prevention, but how were they obtained, preserved, and handled.
3-Line 210: text reads: "....in addiiton to isothermy." Sentence is too telegraphic, and it should explain why is that advantageous.
4-Figure 3 is called after figure 4, and has no explicative legend. Nobody knows what is in the tubes, and on the gel of panels B and C, respectively.
5-Lines 186-187 and Figure 4: The text says "selected as the optimal condition because of the higher specificity and enzyme inhibition at higher temperatures", but it is unclear how that was based on the curves n figure 4, since the 65 degrees curve looks almost the sabe as 66, for example. Some curves looked to be almost the same.
6-Figure 2 requires more clear legend to clarify that the tubes on panel B correspond to the viruses in the lagend of panel A.
7-Legend of figure 5 is utterly improper and unclear. Which positive samples in the curves correspond to the tubes? The comment about disagreement between tests should appear in the Results text, and not thrown in the legend of figure 5.
8-A comment should be made on the cost of this assay as compared to PCR.

Minor issues:
Line 70: the word 'is' is repeated twice.
Table 1, Line 1: Primers for the S gene.
Line 136: measured at 6-s intervals.
Line 160: I presume it should read "positive control plasmid pGEM-S was used along with every tested batch".
Line 182: HS-72 were selected as optimal primers.

---

## Round 0.2 · accepted · Accept

The changes in the revised manuscript were satisfactory.

#